# Sodium-Glucose Cotransporter-2 Inhibitors—Miracle Drugs for the Treatment of Chronic Kidney Disease Irrespective of the Diabetes Status: Lessons from the Dedicated Kidney Disease-Focused CREDENCE and DAPA-CKD Trials

**DOI:** 10.3390/ijms232213749

**Published:** 2022-11-09

**Authors:** Tomohito Gohda, Maki Murakoshi

**Affiliations:** Department of Nephrology, Juntendo University Faculty of Medicine, 2-1-1 Hongo, Bunkyo-ku, Tokyo 113-8421, Japan

**Keywords:** chronic kidney disease, diabetic kidney disease, sodium-glucose cotransporter-2 inhibitor, tumor necrosis factor receptors

## Abstract

Diabetic kidney disease (DKD) is the leading cause of chronic kidney disease (CKD) and end-stage kidney disease worldwide. In Japan, the proportion of new patients requiring dialysis due to DKD has remained unchanged over the past five years. Early diagnosis and treatment are extremely important for the prevention of DKD progression. Albuminuria is the most promising biomarker currently available for diagnosing DKD and predicting its prognosis at an early stage; however, it has relatively poor specificity and sensitivity for DKD. Measuring the serum levels of tumor necrosis factor receptors (TNFRs; TNFR1 and TNFR2) is an alternative for predicting the prognosis of patients with CKD, irrespective of their diabetes status. Cardiorenal risk factor management and renin–angiotensin system inhibitor usage are effective in slowing the DKD progression, although the residual risk remains high in patients with DKD. Recently, two classes of antihyperglycemic agents, sodium–glucose cotransporter 2 (SGLT2) inhibitors and glucagon-like peptide-1 receptor agonists, in addition to nonsteroidal selective mineralocorticoid receptor antagonists, which are less potent blood pressure-lowering and potassium-sparing agents, have emerged as cardiorenal disease-modifying therapies for preventing the DKD progression. This review focused on the SGLT2 inhibitor-based therapeutic strategies that have demonstrated cardiorenal benefits in patients with type 2 diabetes.

## 1. Introduction

Diabetic kidney disease (DKD) is the main cause of chronic kidney disease (CKD) and end-stage kidney disease (ESKD) worldwide. DKD is defined as an estimated glomerular filtration rate (eGFR) of <60 mL/min/1.73 m^2^ and/or the presence of microalbuminuria (i.e., albuminuria > 30 mg/g creatinine or 30 mg/day). In Japan, DKD has been reported to be the most common cause of ESKD since 1998 [1]. The proportion of new patients requiring dialysis, due to DKD has remained unchanged over the past five years [2]. According to the United States Renal Data System Annual Data Report of 2018, Japan has the second highest proportion of patients requiring dialysis, following Taiwan [3].

CKD is an independent predictor of the cardiovascular disease (CVD) and mortality in patients with diabetes [4]. Therefore, early diagnosis and treatment are extremely important for the prevention of not only the DKD progression but also CVD. Albuminuria is the most promising biomarker currently available for diagnosing DKD and predicting its prognosis at an early stage. However, albuminuria has a relatively low specificity and sensitivity for DKD, indicating an urgent clinical need for the development of markers of DKD other than albuminuria [5].

Patients with diabetes require treatments that prevent cardiorenal disease development. Such interventions include intensive blood glucose control to prevent the development of microalbuminuria in patients recently diagnosed with diabetes [6,7]; however, this approach requires careful attention, in terms of hypoglycemia, especially in patients with diabetes of long duration [8]. Dipeptidyl peptidase 4 inhibitors do not cause hypoglycemia, even in patients with DKD; however, they have been reported to have no beneficial effects in patients with CVD [9]. Multifactorial management of cardiorenal risk factors, such as hyperglycemia, dyslipidemia, and hypertension, and the use of angiotensin-converting enzyme inhibitors or angiotensin II type 1 receptor blockers have demonstrated a utility in slowing the progression of DKD [10]. Renin–angiotensin system (RAS) inhibitors, such as losartan and irbesartan, are commonly used drugs for the treatment of DKD that have gained federal regulatory approval in patients with diabetes manifesting albuminuria; however, relatively few patients are prescribed optimal guideline-recommended doses of the RAS inhibitors, partly due to hyperkalemia and hypotension [11]. Even when optimal doses of RAS inhibitors are used, the residual risk for the progression of ESKD in patients with DKD remains high [12,13]. Furthermore, the utility of RAS inhibitors in diabetic patients with normotension and normoalbuminuria remains unclear [14].

Sodium–glucose cotransporter 2 (SGLT2) inhibitors [15,16], nonsteroidal mineralocorticoid receptor antagonists (MRAs) [17], and glucagon-like peptide-1 receptor agonists (GLP-1 RAs) [18] have emerged as cardiorenal disease-modifying therapies for the prevention of the DKD progression. Herein, we discuss treatment strategies in patients with DKD, particularly focusing on the results of the Canagliflozin and Renal Events in Diabetes with Established Nephropathy Clinical Evaluation (CREDENCE) and Dapagliflozin and Prevention of Adverse Outcomes in Chronic Kidney Disease (DAPA-CKD) trials.

## 2. Importance of Early Diagnosis and Therapeutic Intervention in Patients with DKD

As shown in Figure 1, intervening at the time of normal kidney function may prolong the time to the onset of ESKD, compared with intervening after kidney function decline. For example, if the eGFR at diagnosis is 70 mL/min/1.73 m^2^ and the annual eGFR decline rate is −5 mL/min/1.73 m^2^, the patient will develop ESKD (eGFR: 15 mL/min/1.73 m^2^) at an age of 81 years. However, if the therapeutic intervention can halve the annual eGFR decline rate, the age at which ESKD develops can be extended to 92 years. Moreover, if the eGFR at the time of diagnosis is already 30 mL/min/1.73 m^2^, the patient will develop ESKD at an age of 73 years. In this case, even if the same therapeutic intervention is used, the age at which ESKD develops can only be extended to 76 years.

A graph presenting how early and late interventions affect ESKD onset. Early diagnosis and therapeutic intervention in patients with early declines in eGFR can delay further declines in renal function [19]. Late therapeutic intervention after a declined kidney function will reduce the delay period to ESKD (three-year extension from 11-year extension) even if the same prevention of kidney function decline (annual eGFR decline rate: −5 mL/min/1.73 m^2^ from −2.5 mL/min/1.73 m^2^) is achieved as during an early intervention.

## 3. Prognostic Biomarkers of eGFR Decline: Circulating Tumor Necrosis Factor (TNF) Receptors (TNFRs)

Identification of patients at risk of early declines in eGFR is of great clinical importance. Circulating TNFRs (TNFR1 and TNFR2) are potential biomarkers for predicting future eGFR decline in patients with CKD, with or without diabetes [20,21,22,23,24]. Even in diabetic patients with normoalbuminuria, TNFRs, but not eGFR or albuminuria, have been reported to predict composite kidney outcomes, including 40% reductions in eGFR from the baseline; ESKD; and death from renal causes after adjustment for the baseline eGFR levels, albuminuria, and glycated hemoglobin (HbA1c) [25]. Moreover, TNFRs aid in predicting all-cause mortality in patients on hemodialysis, irrespective of their diabetes status [26]. A recent post-hoc analysis of large-scale randomized controlled trials (RCTs), including the Canagliflozin Cardiovascular Assessment Study (CANVAS), African American Study of Kidney Disease and Hypertension (AASK), and Veterans Affairs Nephropathy in Diabetes trial (VA NEPHRON-D), demonstrated that changes in the TNFR levels also predict the eGFR decline, indicating the potential utility of TNFRs as alternatives to albuminuria as biomarkers in therapeutic monitoring and prognosis of CKD [27,28].

## 4. Effects of the SGLT2 Inhibitors on Kidney Outcomes in Patients with Nondiabetic CKD or Diabetes

The results of previous cardiovascular outcome trials (CVOTs) indicated that SGLT2 inhibitors improve kidney outcomes in patients with type 2 diabetes (T2D) and normal renal function, thereby leading to the initiation of multiple trials on kidney outcomes that enrolled patients with advanced CKD (CREDENCE, DAPA-CKD, and the Study of Heart and Kidney Protection with Empagliflozin [EMPA-KIDNEY]) [15,16]. As shown in Table 1, the clinical characteristics of the enrolled patients, including age, sex, body mass index, and urinary albumin-to-creatinine ratio, were comparable between the CREDENCE and DAPA-CKD trials. The inclusion criteria for the CREDENCE trial were different from those for the DAPA-CKD trial, in terms of the baseline eGFR (CREDENCE: 30 to 90 mL/min/1.73 m^2^; DAPA-CKD: 25 to 75 mL/min/1.73 m^2^). Compared with the CREDENCE trial, the baseline eGFR was low and stage 4, CKD was frequent in the DAPA-CKD trial. The CREDENCE trial was the first RCT to evaluate the effects of the SGLT2 inhibitors on primary kidney outcomes in patients with T2D and macroalbuminuria (i.e., classical diabetic nephropathy) [15]. Moreover, the DAPA-CKD trial was the first kidney outcome study to evaluate the effects of the SGLT2 inhibitors on patients with CKD with and without T2D (patients with diabetes, 68%; patients without diabetes, 32%) [16]. The abovementioned two trials reported significant reductions in composite kidney outcomes, including the doubling of serum creatinine, 50% reductions in eGFR from the baseline, ESKD, and death from renal causes or CVD (CREDENCE, 30%; DAPA-CKD, 39%).

In the DAPA-CKD trial, the effects of the SGLT2 inhibitors on kidney outcomes did not change in patients with CKD, regardless of their diabetes status [16]. In patients with CKD with normoglycemia or prediabetes, dapagliflozin did not decrease HbA1c, compared with the placebo during follow-up. In patients with CKD with diabetes whose mean eGFR was 43.1 mL/min/1.73 m^2^, dapagliflozin only decreased HbA1c by 0.1% [29]. Moreover, the mediation analysis performed using the CANVAS program demonstrated that the serum glucose-lowering effects of HbA1c did not contribute to the kidney outcomes [30]. Thus, the SGLT2 inhibitors are effective in reducing the cardiorenal risk, independent of their glucose-lowering effects. Indeed, even in patients with relatively good blood glucose control (i.e., HbA1c < 7%), administration of the SGLT2 inhibitors improved the cardiorenal outcomes in diabetic patients with macroalbuminuria [31]. Therefore, the addition of the SGLT2 inhibitors without individualized target HbA1c levels should be considered in patients with T2D.

In the Dapagliflozin Effect on Cardiovascular Events–Thrombolysis in Myocardial Infarction 58 (DECLARE-TIMI 58) trial, dapagliflozin reduced the kidney outcomes with no evidence of heterogeneity in the levels of the baseline systolic blood pressure (BP) in patients with T2D [32]. Even in diabetic patients with normotension (i.e., systolic BP < 120 mmHg), dapagliflozin reduced the kidney outcomes. Of note, there was no difference in the incidence of volume depletion or acute kidney injury (AKI) at any level of systolic BP with dapagliflozin. The benefit of dapagliflozin, compared with the placebo in patients with CKD was consistent, regardless of the dose of the RAS inhibitors taken at the baseline [33]. Thus, the SGLT2 inhibitors may be effective in patients with CKD with normotension receiving or not receiving RAS inhibitors.

## 5. Renoprotective Effects of the SGLT2 Inhibitors beyond Reductions in Albuminuria

In both the CREDENCE and DAPA-CKD trials and the previous CVOTs [34,35], the SGLT2 inhibitors have been reported to decrease albuminuria by approximately 30% in patients with T2D (Table 1) [36,37,38]. Of note, the effect of the SGLT2 inhibitors on albuminuria was greater in patients with diabetes than in patients without diabetes and in patients with diabetic nephropathy than in patients with other etiologies of CKD [34]. However, this effect varied among individual patients. Heerspink et al. [39] reported that dapagliflozin did not decrease albuminuria during the first six months of treatment in approximately 30% of the patients in a pooled analysis of 11 phase 3 RCTs, comprising diabetic patients with a high prevalence of microalbuminuria (83%). A similar finding was observed for diabetic patients with macroalbuminuria. In the CREDENCE trial, albuminuria increased in approximately 30% of the diabetic patients with macroalbuminuria during the first six months of treatment with canagliflozin [40]. Despite no change in albuminuria, dapagliflozin slowed down the decrease in eGFR, compared with the placebo in the DAPA-CKD trial, possibly due to the renoprotective effects apart from reducing albuminuria [34]. In the CREDENCE trial, a 30% reduction in albuminuria, during the early stages of treatment with canagliflozin, was strongly associated with improved kidney outcomes, compared with the placebo, even if this reduction was comparable between the canagliflozin-treated and placebo groups, indicating that canagliflozin has renoprotective effects other than reducing albuminuria [40]. Mediation analysis performed using the CANVAS program demonstrated the renoprotective effects of canagliflozin, including a reduction in BP, reduction in the serum levels of uric acid, and improvements in anemia [30].

## 6. Initial eGFR Declines and Volume Depletion: Concerning Side-Effects of the SGLT2 Inhibitors

SGLT2 inhibitors augment the tubuloglomerular feedback by increasing sodium and chloride delivery to the macula densa, thereby leading to acute initial declines in eGFR, resulting from the reductions in the intraglomerular pressure. This issue has increased hesitancy of clinicians to administer SGLT2 inhibitors to patients with a decreased kidney function. In the CREDENCE and DAPA-CKD trials, the proportion of patients with an initial decline in eGFR (>10%) was 45% and 49%, respectively, for the SGLT2 inhibitor treatment; an approximately three-fold higher odds ratio was obtained with the SGLT2 inhibitor than with the placebo in both the trials [41,42]. The common clinical characteristics of patients with an acute initial decline in eGFR in these trials were an older age, longer duration of diabetes, high systolic BP, and use of diuretics [41,42,43]. In the DAPA-CKD trial, the renal benefits of dapagliflozin for long-term eGFR trajectory, differed across various categories of acute initial declines in eGFR (>10%, 0–≤10%, ≤0%) [42]; however, in the CREDENCE trial, long-term eGFR trajectories were similar regardless of the extent of the acute initial declines in eGFR [41]. These discordant results were apparently derived from baseline eGFRs. In fact, within the subgroup of patients in the CREDENCE trial (eGFR, 30–45 mL/min/1.73 m^2^) whose baseline eGFRs were comparable with those of patients in the DAPA-CKD trial (eGFR, 43 mL/min/1.73 m^2^), long-term eGFR trajectory was improved in patients with >10% acute initial declines in eGFR but not in those with modest or no acute declines in eGFR. A further study is needed to investigate whether >10% acute initial declines in eGFR after initiation of the SGLT2 inhibitors has a beneficial effect on long-term eGFR trajectories. Moreover, in the Dapagliflozin and Prevention of Adverse Outcomes in Heart Failure (DAPA-HF) trial, >10% acute declines in eGFR after the initiation of the SGLT2 inhibitors in patients with heart failure (HF) with reduced ejection fraction (HFrEF), showed better composite outcomes of worsening HF and CV deaths than no decline in eGFR [44]. In real-world clinical settings and practice, the continuous use of SGLT2 inhibitors, compared with the discontinuation at six months in patients with diabetes, newly introduced to the SGLT2 inhibitors is associated with a reduced risk of CV and kidney outcomes, irrespective of the acute declines in eGFR [45].

A further concern regarding the use of the SGLT2 inhibitors, is volume depletion. Although volume depletion is a relatively common adverse event associated with the use of SGLT2 inhibitors (approximately 5%), the incidence of AKI was low in both kidney outcome trials (CREDENCE and DAPA-CKD) likely due to: (1) reductions in tubular ischemia through the attenuation of sodium and glucose reabsorption; (2) improvements in anemia, leading to an improved renal oxygenation; and (3) reductions in the usage of diuretics. In the CREDENCE trial, canagliflozin decreased the risk of adverse renal events by approximately 30%, in comparison with the placebo [46]. However, during the first year, the increase in the incidence of these adverse events was greater in the canagliflozin-treated group than that in the placebo group, indicating that the investigators may have misidentified acute initial declines in eGFR as adverse events, as Kaplan–Meier curves crossed at 12 months and then continued to diverge until the end of the trial. Interestingly, patients treated with canagliflozin were approximately twice as likely to recover full kidney function after AKI onset, compared with the placebo. In the DAPA-CKD trial, dapagliflozin decreased the risk of investigator-reported AKI by approximately 30%, compared with the placebo [47]. The effect of dapagliflozin exhibited no heterogeneity in comparison with the placebo, in terms of the abrupt decline in kidney function among subgroups, such as patients with and without T2D, those receiving diuretics, and those with different levels of baseline albuminuria (1 g/gCr or more/less) and different eGFRs (45 mL/min/1.73 m^2^ or more/less). Of note, the risk of ESKD in patients who develop AKI during the study period dramatically increased more than 10-fold, compared with those who did not. A meta-analysis of the data from three major CVOTs (the Empagliflozin Cardiovascular Outcome Event Trial in Type 2 Diabetes Mellitus Patients—Removing Excess Glucose [EMPA-REG OUTCOME], CANVAS, and DECLARE-TIMI 58) with SGLT2 inhibitors in patients with T2D, revealed a robust reduction in the risk of AKI by 34%; however, these events were nonadjudicated [48].

## 7. Hyperkalemia

A decline in kidney function leads to hyperkalemia, due to the decreased potassium excretion from the kidneys. In addition, diabetes, HF, and the use of RAS inhibitors and MRAs are associated with an increased risk of hyperkalemia. Hyperkalemia, in turn, is an independent risk factor for mortality and CV events [49]. In the Reduction of Endpoints in noninsulin dependent diabetes mellitus with the Angiotensin II Antagonist Losartan (RENAAL) trial, patients treated with losartan were 2.8 times more likely to develop hyperkalemia (≥5.0 mmol/L) at six months than those not treated with losartan, and hyperkalemia (≥5.0 mmol/L) at six months were associated with a 22% higher risk of doubling of the serum creatinine level or for ESKD, after the adjustment for relevant factors [50]. Therefore, management of hyperkalemia is profoundly important in patients with DKD. The meta-analysis of data from four CVOTs and two trials of kidney outcomes with SGLT2 inhibitors in patients with T2D or CKD showed a significant reduction in the risk of serious hyperkalemia (≥6.0 mmol/L) by 16% without any heterogeneity across the trials [51]. The reduction in the risk of serious hyperkalemia remained unchanged when the data from the DAPA-HF and the Empagliflozin Outcome Trial in Patients with Heart Failure with Reduced Ejection Fraction (EMPEROR-Reduced) studies on patients with and without T2D with HFrEF were added into the analysis [51]. The effect also remained unchanged, regardless of the use of MRAs at the baseline. A randomized crossover trial of dapagliflozin, MRA eplerenone, and their combination in patients with CKD (Rotation for Optimal Targeting of Albuminuria and Treatment Evaluation-3), demonstrated that the adjunct combination therapy with RAS inhibitors resulted in not only a robust reduction in the incidence of albuminuria but also hyperkalemia, compared with eplerenone therapy alone (4.3% versus 17.4%; *p* < 0.01) [52]. In the Finerenone in Reducing Kidney Failure and Disease Progression in Diabetic Kidney Disease (FIDELIO-DKD) trial, finerenone, a novel, nonsteroidal, selective MRA, reduced the urine albumin-to-creatinine ratio in patients with DKD, regardless of whether the patients received SGLT2 inhibitors or GLP-1 RA at the baseline, with no heterogeneity [53,54]. However, the data regarding the renoprotective effects of these combination therapies and their safety in patients with hyperkalemia are lacking. In the CONFIDENCE trial, researchers are investigating whether the combination of finerenone and the SGLT2 inhibitor empagliflozin is superior to either drug alone, in reducing the degree of albuminuria and the risk of hyperkalemia (NCT05254002) [55].

## 8. Effects of the SGLT2 Inhibitors in Patients with CKD and Normoalbuinuria

The data regarding the renoprotective effects of SGLT2 inhibitors in patients with CKD and normoalbuminuria are lacking. The EMPA-KIDNEY trial was stopped earlier than planned, due to the demonstrated efficacy in kidney outcomes, suggesting that SGLT2 inhibitors may expand indications for patients with CKD without albuminuria. Indeed, CVOTs in patients with T2D, also demonstrated that the renoprotective effects of SGLT2 inhibitors are independent of the amount of albuminuria [56]. The pleiotropic effects of the SGLT2 inhibitors, such as the improvements in tubulointerstitial ischemia, inflammation, and mitochondrial function, may contribute to improved kidney outcomes in patients with diabetes or CKD with normoalbuminuria [57,58].

## 9. Effects of the SGLT2 Inhibitors on Atherosclerotic CVD (ASCVD) and HF in Patients with CKD

In CVOTs, evaluating the effects of the SGLT2 inhibitors in patients with T2D, the effect of the SGLT2 inhibitors on 3-point major adverse events (3P-MACE) was observed only in the secondary prevention group [59]; however, the CREDENCE trial demonstrated that the risk of 3P-MACE was reduced even in the primary prevention group [60]. These discordant results appear to be derived from the event rates of 3P-MACE in each trial (DECLARE-TIMI 58, CANVAS program, EMPA-REG OUTCOME, and CREDENCE) [61], which appear to be chiefly attributable to kidney function and the rates of prior ASCVD, as shown in Table 2. In the DAPA-CKD trial, dapagliflozin did not reduce the risk of 3P-MACE, likely due to a low prevalence of prior ASCVD (37%) and partial inclusion of patients with nondiabetic CKD (33%). In the CREDENCE trial, the largest relative risk reductions (RRRs) of CVD were observed in patients with T2D, compared with those observed in CVOTs (DECLARE-TIMI 58, CANVAS program, and EMPA-REG OUTCOME) [61]. Moreover, among the four trials (DECLARE-TIMI 58, CANVAS program, EMPA-REG OUTCOME, and CREDENCE), the CREDENCE trial showed the lowest RRRs of the composite kidney outcomes [61]. Kidney function reserve, which defines the peak eGFR induced via the stress response minus the baseline eGFR, was reported to be negatively associated with eGFR [62]. Kluger et al. stated that the higher the kidney function reserve, the greater is the positive impact of the SGLT2 inhibitors on kidney outcomes [61].

In both the CREDENCE and DAPA-CKD trials, the SGLT2 inhibitors reduced the risk of CV death or hospitalization, due to HF by approximately 30%, in patients with CKD, irrespective of the presence or absence of prior ASCVD [60,63]. This finding was consistent with that of previous CVOTs. In the dedicated HF-focused EMPEROR-Reduced and Empagliflozin Outcome Trial in Patients with Chronic Heart failure with Preserved Ejection Fraction (EMPEROR-Preserved) [64,65], SGLT2 inhibitors reduced kidney outcomes and the risk of hospitalization, due to HF in patients with both HF with preserved ejection (HFpEF) and HFrEF. However, the effects of the SGLT2 inhibitors on hospitalization, due to HF and kidney outcomes appear to be dependent on the ejection fraction in patients with HFpEF, indicating that the effects of the SGLT2 inhibitors were reduced in patients with greater ejection fractions [66].

## 10. Promising Therapeutic Strategy

Early diagnosis and intervention are of utmost clinical importance in DKD. In addition to eGFR and albuminuria, circulating TNFRs have the potential utility in identifying patients at risk of progressive declines in kidney function. Current clinical management of patients with diabetes comprises the correction of blood glucose, BP, and dyslipidemia with lifestyle modifications, such as appropriate exercise, diet, and cessation of smoking. The goal of treatment, particularly in patients with DKD, is the reduction in albuminuria and the preservation of kidney function while preventing CV events. However, patients with diabetes exhibit an increased prevalence of comorbidities and complications. Many previous clinical studies, mainly in patients with diabetes, have added to the existing knowledge on the risk factors and efficacious interventions for DKD and enabled the development of personalized treatment strategies that consider the comorbidities and complications in individual patients with diabetes, to reduce the residual risk of DKD. However, there remains an urgent clinical need to reduce the ever increasing number of patients with DKD using a combination of treatments, including RAS inhibitors, SGLT2 inhibitors, MRAs, and GLP-1 RAs, which are currently the four mainstay treatments for DKD (Figure 2).

Patients with diabetes are at an increased risk of comorbidities, such as hypertension, hyperuricemia, and obesity, as well as complications, such as CKD, ASCVD, and HF. RAS inhibitors are highly recommended in hypertensive patients; however, the other three treatment agents also exhibit mild blood pressure-lowering effects (approximately 2–3 mmHg). SGLT2 inhibitors and some RAS inhibitors, such as losartan and irbesartan, exhibit mild uric acid-lowering effects. SGLT2 inhibitors and GLP-1 RAs exhibit body weight-lowering effects that differ from those of other antihyperglycemic agents. RAS inhibitors, SGLT2 inhibitors, and MRAs can be used for preventing CKD (i.e., worsening kidney function decline and elevated albuminuria). Furthermore, GLP-1 RAs exhibit antialbuminuria effects, which are comparable to those of the other three agents, particularly in patients with diabetes and macroalbuminuria. Anemia and hyperkalemia are common complications observed in patients with advanced CKD. SGLT2 inhibitors exhibit potassium-lowering effects and have been shown to improve anemia.

The abovementioned four therapeutic agents (RAS inhibitors, SGT2 inhibitors, MRAs, and GLP-1 RAs) have a demonstrated efficacy in preventing the development of ASCVD. Except GLP-1 RAs, the other three agents have a utility in the prevention of hospitalization, due to HF. GLP-1 RAs, but not SGLT2 inhibitors, have a utility in reducing the risk of cerebral infarction. Patients with prior ASCVD or HF are often hypotensive. In such cases, the use of RAS inhibitors is more challenging, due to reduced renal blood flow. Thus, patients with diabetes should receive personalized treatment that considers individual comorbidities and complications to reduce the residual risk of cardiorenal disease.

## 11. Conclusions

SGLT2 inhibitors not only lower the blood glucose by inhibiting glucose reabsorption in the proximal tubule but also prevent kidney and HF outcomes through various pleiotropic actions. Especially in patients with impaired kidney function, SGLT2 inhibitors are more of a cardiorenal risk-reducing agent than a hypoglycemic agent. It is hoped that the SGLT2 inhibitors will be appropriately administered to as many patients at a high risk of cardiorenal disease as possible.

## Figures and Tables

**Figure 1 ijms-23-13749-f001:**
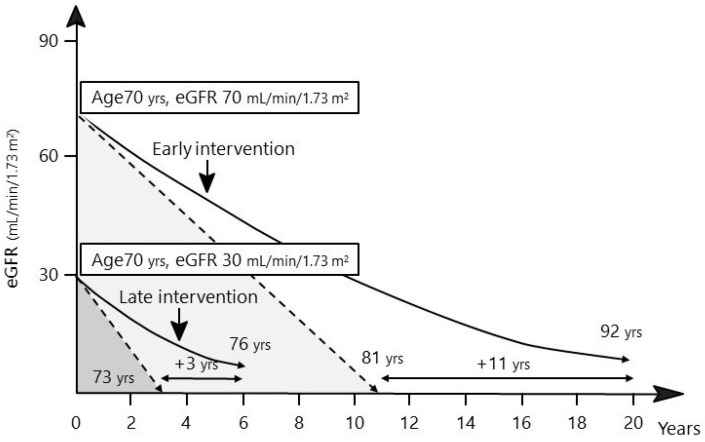
Importance of an early diagnosis and therapeutic intervention in patients with DKD.

**Figure 2 ijms-23-13749-f002:**
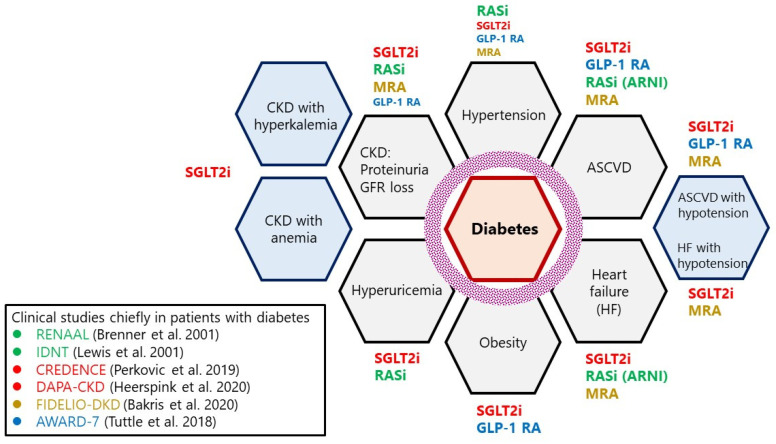
Four treatment drugs for DKD (RAS inhibitors, SGT2 inhibitors, MRAs, and GLP-1 RAs). Clinical studies chiefly in patients with diabetes, RENAAL [12], IDNT [13], CREDENCE [15], DAPA-CKD [16], FIDELIO-DKD [17], AWARD-7 [18].

**Table 1 ijms-23-13749-t001:** Baseline characteristics and cardio-renal outcomes of the enrolled patients in each trial.

	CREDENCE [15] (Canagliflozin)	DAPA-CKD [16] (Dapagliflozin)	DAPA-CKD (with T2D)	DAPA-CKD (without T2D)
n	4401	4304	2906	1398
Age (year)	63	62	64	56
Male sex (%)	66	67	67	67
Body mass index	31.3	29.5	30.3	27.9
T2D (%)	100	68	100	0
HbA1c (%)	8.3	7.1	7.8	5.6
LSM change in HbA1c: % (95% CI)	−0.3 (−0.2 to −0.3)	−0.1 (−0.1 to 0.0)	−0.1 (−0.2 to 0.0)	0.0 (−0.2 to 0.2)
ACR (mg/g)	927	949	1017	861
Geometric mean ACR change: % (95% CI)	−31 (−35 to −26)	−29 (−33 to −25)	−35 (−39 to −31)	−15 (−23 to −6)
Stage 4 CKD (%)	4	14	14	16
eGFR (mL/min/1.73 m^2^)	56	43	44	42
LSM change in the eGFR slope (mL/min/1.73 m^2^/year)	1.52 (1.11 to 1.93)	0.93 (0.61 to 1.25)	1.18 (0.79 to 1.56)	0.46 (−0.10 to 1.03)
Initial decline in eGFR * (SGLT2i vs. PBO) (mL/min/1.73 m^2^)	−3.7 vs. −0.6	−4.0 vs. −0.8	−3.2 vs. −0.6	−2.8 vs. −0.8
Decline in eGFR after initial phase (SGLT2i vs. PBO) (mL/min/1.73 m^2^/year)	−1.9 vs. −4.6	−1.7 vs. −3.6	−1.6 vs. −3.8	−1.9 vs. −3.2
Prior CV disease (%)	50	37	44	24
Primary endpoint **: HR (95% CI)	0.70 (0.59 to 0.82)	0.61 (0.51 to 0.72)	0.64 (0.52 to 0.79)	0.50 (0.35 to 0.72)
CV death or HHF ***: HR (95% CI)	0.69 (0.57 to 0.83)	0.71 (0.55 to 0.92)	0.70 (0.53 to 0.92)	0.79 (0.40 to 1.55)
All cause mortality ***: HR (95% CI)	0.83 (0.68 to 1.02)	0.69 (0.53 to 0.88)	0.74 (0.56 to 0.98)	0.52 (0.29 to 0.93)

ACR, Urinary albumin to creatinine ratio; CKD, chronic kidney disease; CI, confidence interval; CV, cardiovascular; eGFR, estimated glomerular filtration rate; ESKD, end stage kidney disease; HbA1c, glycated hemoglobin; HHF, hospital admission for heart failure; HR, hazard ratio; LSM, least square mean; NA, not available; PBO, placebo; SGLT2i, sodium-glucose cotransporter 2 inhibitors; T2D, type 2 diabetes. * CREDENCE: First 3 weeks; DAPA-CKD, First 2 weeks. ** Primary endpoint: CREDENCE, 57% reduction in eGFR from baseline, ESKD, and death from renal or CV disease causes; DAPA-CKD, 50% reduction in eGFR from baseline, ESKD, renal death, CV death *** Secondary endpoint.

**Table 2 ijms-23-13749-t002:** Baseline characteristics and cardio-renal outcomes of the enrolled patients in each study.

	DECLARE-TIMI58	CANVAS	EMPA-REG OUTCOME	CREDENCE	DAPA-CKD
Prior CVD (%)	40.6	65.6	99.2	50.4	37.4
Baseline eGFR (mL/min/1.73 m^2^)	85	77	74	56	43
Composite kidney outcomes *: HR (95% CI)	0.53 (0.43 to 0.66)	0.60 (0.47 to 0.77)	0.54 (0.40 to 0.75)	0.70 (0.59 to 0.82)	0.61 (0.51 to 0.72)
3P-MACE: HR (95% CI)	0.93 (0.84 to 1.03)	0.86 (0.75 to 0.97)	0.86 (0.74 to 0.99)	0.80 (0.67 to 0.95)	0.92 (0.72 to 1.16)
HHF: HR (95% CI)	0.73 (0.61 to 0.88)	0.67 (0.52 to 0.87)	0.65 (0.50 to 0.85)	0.61 (0.47 to 0.80)	0.51 (0.34 to 0.76)
HHF and CV death: HR (95% CI)	0.85 (0.73 to 0.95)	0.78 (0.67 to 0.91)	0.66 (0.55 to 0.79)	0.69 (0.57 to 0.83)	0.71 (0.55 to 0.92)

Abbreviations used in this table are the same as in Table 1. * DECLARE-TIMI 58: ≥40% reduction in eGFR, ESKD (chronic dialysis, transplant or sustained eGFR < 15), or renal/CV death; CANVAS: ≥40% reduction in eGFR, ESKD or renal death; EMPA–REG OUTCOME: doubling of serum creatinine (Cr) with eGFR ≤ 45, ESKD, or renal death; CREDENCE: doubling of serum Cr, ESKD, renal/CV death; DAPA-CKD: ≥50% reduction in eGFR, ESKD, renal/CV death.

## Data Availability

Not applicable.

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
