# Peer review of "Sodium-Glucose Cotransporter-2 Inhibitors—Miracle Drugs for the Treatment of Chronic Kidney Disease Irrespective of the Diabetes Status: Lessons from the Dedicated Kidney Disease-Focused CREDENCE and DAPA-CKD Trials"

_ijms, 2022, doi:10.3390/ijms232213749_

Round 1

Reviewer 1 Report

The review with title: SGLT2 Inhibitors—Miracle Drugs for the Treatment of CKD Irrespective of Diabetes Status: Lessons from the Dedicated Kidney Disease-focused CREDENCE and  DAPA-CKD Trials” provides a review regarding SGLT2 inhibitor-based therapies for diabetic kidney disease. The authors analyse and compare the results of different clinical trials in order to identify the appropriate treatments for diabetic patients suffering from different comorbidities and complications in an approach for personalized medicine.

However, I found the text difficult to read, with sentences that lacked coherency. Furthermore, some sections had contradictory messages which were not resolved by the authors, and which therefore do not bring anything new from what has already been published by the authors of the corresponding studies.

This review also does not provide any novel information from the authors, meaning suggestions or personal opinion as experts in the field, but they just mention results and compare studies that have already been published.

Besides, there are no summary tables with the conclusions of each study. Just a minor discussion is found on the conclusion section but it is also complicated to read. Since this is basically a comparison between few clinical trials, I strongly believe that some tables must had been included in order to mention each study, with the corresponding drugs and their effects on the distinct individuals. All this information is spread all over the text and it is not easy to made a conclusion out of it.

Besides, I also found some other issues that should be addressed:

Line 15. There is a “the” that should be eliminated before “sensitivity”.

Line 41 to 43. It does not have sense. Why do you compare the risk of mortality between CV and DKD to justify the relevance of early diagnosis? You should explain here the interconnection between them in cardio-renal diseases.

Line 47. There is a “the” that should be eliminated before “sensitivity”.

Line 50. “For the prevention” instead of “of the prevention”.

Line 51,52 and 53. You state “Such treatment includes intensive blood glucose control” but this is not a treatment, it is a control.

Line 51 to 55. This sentence has no sense. Glucose control is not a therapeutic approach, is just a control so no clear why you state that careful attention needs to be taken in hypoglycemia patients. Please clarify this sentence.

Line 63. What do you mean with the residual risk? Not explained.

Line 76 and 77. I don’t understand the sentences. It is explained with an example in the subsequent sequences, but the initial one has to be reformulated appropriately.

Line 94-96. This is quite redundant; it has already been mentioned in the abstract and introduction.

Line 195: What do you mean with real world data?

Line 196: Sentence is incomplete. The initiation of what? Of the treatment?

Line 197: Continued used of SLGT2 for how long? What does imply a continued used?

On section 2 you mention the relevance of early diagnosis based on the levels of GFR, where lower GFR levels are associated with worse prognosis. However, in section 7, where you describe the beneficial effects of SLGT2, you mentioned that treatment with SLGT2 is associated with a decline of GFR, but still reduces the risk of CV. How do you explain this discrepancy? This should be explained in the main text.

Line 242. What kind of sensitivity analysis was performed?

Line 267: You state “The results of the EMPA-Kidney trial, which was recently stopped early for efficacy” and then you sate “are expected to expand indications for the use of  SGLT2 inhibitors” . What do you mean with it was stopped for efficacy? If it was stopped how results are expected to bring new light in the use of SGLT2 inhibitors? Please clarify this sentence.

Altogether, I believe it was interesting to make a review about this special topic, however I do not consider the way it has been presented is useful for potential readers interested in the field.

I suggest that strong review should be performed, with more comprehensive sentences, with explanatory and summarizing tables, additional figures, etc. that make a clear point of the status and the conclusions provided by this review based on several clinical studies.

Author Response

Reviewer #1

Line 15- “specificity and the” remove the “the” before sensitivity
Thank you for pointing out this error. We have corrected it accordingly. 

Line 41 to 43. It does not have sense. Why do you compare the risk of mortality between CV and DKD to justify the relevance of early diagnosis? You should explain here the interconnection between them in cardio-renal diseases.

We have revised the introduction in line with your suggestion.

Line 47. There is a “the” that should be eliminated before “sensitivity”.

Line 50. “For the prevention” instead of “of the prevention”.

Line 51,52 and 53. You state “Such treatment includes intensive blood glucose control” but this is not a treatment, it is a control.

Line 63. What do you mean with the residual risk? Not explained.

Line 76 and 77. I don’t understand the sentences. It is explained with an example in the subsequent sequences, but the initial one has to be reformulated appropriately.

Thank you for pointing out these errors. We have corrected these accordingly.

Line 94-96. This is quite redundant; it has already been mentioned in the abstract and introduction.

We have deleted this sentence according to your suggestion.

Line 195: What do you mean with real world data?

Line 196: Sentence is incomplete. The initiation of what? Of the treatment?

Line 197: Continued used of SLGT2 for how long? What does imply a continued used?

We have revised this sentence for better clarity.

On section 2 you mention the relevance of early diagnosis based on the levels of GFR, where lower GFR levels are associated with worse prognosis. However, in section 7, where you describe the beneficial effects of SLGT2, you mentioned that treatment with SLGT2 is associated with a decline of GFR, but still reduces the risk of CV. How do you explain this discrepancy? This should be explained in the main text.

We have revised the manuscript to add this detail according to your suggestion (line 221-230, 323-332).

Line 242. What kind of sensitivity analysis was performed?

We have included this information in the revised manuscript.

Line 267: You state “The results of the EMPA-Kidney trial, which was recently stopped early for efficacy” and then you sate “are expected to expand indications for the use of  SGLT2 inhibitors” . What do you mean with it was stopped for efficacy? If it was stopped how results are expected to bring new light in the use of SGLT2 inhibitors? Please clarify this sentence.

We have revised these sentences for better clarity.

Altogether, I believe it was interesting to make a review about this special topic, however I do not consider the way it has been presented is useful for potential readers interested in the field.

I suggest that strong review should be performed, with more comprehensive sentences, with explanatory and summarizing tables, additional figures, etc. that make a clear point of the status and the conclusions provided by this review based on several clinical studies.

The revised manuscript has been carefully reviewed by an experienced editor whose first language is English and who specializes in editing papers written by scientists whose native language is not English. We have added tables that summarize each study.

Reviewer 2 Report

Well written article

Author Response

Reviewer #2

Thank you for dedicating your valuable time and efforts in reviewing our manuscript.

Reviewer 3 Report

See attachment

Author Response

Reviewer #3

Abstract

Line 15- “specificity and the” remove the “the” before sensitivity
Thank you for pointing out this error. We have corrected it accordingly.

  1. Introduction

Line 38- put a reference at the end of the sentence.

We have added a relevant reference at this instance.

Line 53-55- the phrase “although this therapeutic approach requires careful attention in terms of hypoglycemia in patients with a longer duration of diabetes” is not clear”. Please rephrase. What do you mean by duration of diabetes?

We have revised this sentence for better clarity.

You can mention some anti-diabetic drugs that have lower risks of hypoglycaemia e.g. the DPP-4 inhibitors, biguanides etc.

We have described that DPP-4 inhibitors are a low-risk type of drugs for hypoglycemia.

Line 58- You can mention examples of the RAS inhibitors

We have added examples of RAS inhibitors.

Line 62-63. This sentence negates the one in line 56-58 (and the use of angiotensin-converting

enzyme inhibitors or angiotensin II type 1 receptor blockers have demonstrated utility in slowing the progression of DKD). Harmonize these sentences.

Thank you for pointing out this error. We have corrected it

accordingly.

  1. Importance of early diagnosis and therapeutic intervention in DKD

Line 76- why underline Figure 1?

Line 86- remove the Figure 1 on the figure. It is already placed at the footnote of the figure.

Thank you for pointing out this error. We have corrected it accordingly.

  1. Prognostic biomarkers of GFR decline circulating TNF receptors

Line 90-91- insert a reference here

We have added relevant references at this instance.

Line 99 – use with or without

Thank you for pointing out this error. We have corrected it accordingly.

  1. Effects of SGLT2 inhibitors on kidney outcomes in patients with non-diabetic CKD or diabetes

Line 115-117- insert a reference.

We have added relevant references at this instance.

Line 136- use “independent” instead of “independently”

Thank you for pointing out this error. We have corrected it accordingly.

  1. Initial estimated GFR declines and volume depletion: concerning side-effects of SGLT2

inhibitors?

Line 197- which antihyperglycemics? Give examples.

We have revised the sentence accordingly.

  1. Hyperkalemia

Line 232- Write NIDDM in full for the first time. Do same for other acronyms.

Thank you for pointing out this error. We have corrected it accordingly.

  1. Conclusions

The conclusion is too long. Move part of the conclusion (figure 2 inclusive) to previous sections. The conclusion should be in one paragraph.

We have revised the conclusion as suggested.

General comment

SGLT2 inhibitors should be defined in terms of their mechanism of action.

All acronyms must be first written in full before subsequent usage

We have revised the manuscript according to your suggestion.

Round 2

Reviewer 1 Report

I would accept the manuscript after all the corrections they have made.